# Targeting of the ELR+CXCL/CXCR1/2 Pathway Is a Relevant Strategy for the Treatment of Paediatric Medulloblastomas

**DOI:** 10.3390/cells11233933

**Published:** 2022-12-05

**Authors:** Manon Penco-Campillo, Clément Molina, Patricia Piris, Nouha Soufi, Manon Carré, Marina Pagnuzzi-Boncompagni, Vincent Picco, Maeva Dufies, Cyril Ronco, Rachid Benhida, Sonia Martial, Gilles Pagès

**Affiliations:** 1Institute for Research on Cancer and Aging (IRCAN), Université Côte d’Azur, CNRS UMR 7284 and INSERM U1081, 33 Avenue de Valombrose, 06107 Nice, France; 2Centre de Recherche en Cancérologie de Marseille (CRCM), Institut Paoli Calmettes, Aix-Marseille Université, Inserm U1068, CNRS UMR 758, 27 Boulevard Jean Moulin, 13273 Marseille, France; 3Centre Scientifique de Monaco (CSM), Biomedical Department, 98000 Monaco, Monaco; 4Roca Therapeutics, 06000 Nice, France; 5Institut de Chimie de Nice UMR 7272, Université Côte d’Azur, Centre National de Recherche Scientifique (CNRS), 06108 Nice, France

**Keywords:** paediatric medulloblastoma, CXCR1/2, ELR+CXCL cytokines, targeted therapy, angiogenesis

## Abstract

Medulloblastoma (MB) is the most common and aggressive paediatric brain tumour. Although the cure rate can be as high as 70%, current treatments (surgery, radio- and chemotherapy) excessively affect the patients’ quality of life. Relapses cannot be controlled by conventional or targeted treatments and are usually fatal. The strong heterogeneity of the disease (four subgroups and several subtypes) is related to innate or acquired resistance to reference treatments. Therefore, more efficient and less-toxic therapies are needed. Here, we demonstrated the efficacy of a novel inhibitor (C29) of CXCR1/2 receptors for ELR+CXCL cytokines for the treatment of childhood MB. The correlation between ELR+CXCL/CXCR1/2 expression and patient survival was determined using the R2: Genomics Analysis and Visualization platform. In vitro efficacy of C29 was evaluated by its ability to inhibit proliferation, migration, invasion, and pseudo-vessel formation of MB cell lines sensitive or resistant to radiotherapy. The growth of experimental MB obtained by MB spheroids on organotypic mouse cerebellar slices was also assayed. ELR+CXCL/CXCR1/2 levels correlated with shorter survival. C29 inhibited proliferation, clone formation, CXCL8/CXCR1/2-dependent migration, invasion, and pseudo-vessel formation by sensitive and radioresistant MB cells. C29 reduced experimental growth of MB in the ex vivo organotypic mouse model and crossed the blood–brain barrier. Targeting CXCR1/2 represents a promising therapeutic strategy for the treatment of paediatric MB in first-line treatment or after relapse following conventional therapy.

## 1. Research in Context

### 1.1. Evidence before This Study

Medulloblastoma (MB), which arises in the cerebellum, is the most common and one of the most aggressive paediatric brain tumours. Despite aggressive multimodal treatment combining surgery, radio- and chemotherapy, associated with considerable side effects [1,2], 30% of patients develop a resistance and relapse with metastases. The patient outcome, nevertheless, is highly variable. It depends on the genetic subgroup identified, the stage of the disease, but especially the metastatic status at the time of diagnosis [3,4]. Despite the improvement of various therapies (anti-angiogenic, anti-inflammatory, anti-immune checkpoints), the management of MB remains delicate when relapses occur, and treatment options are limited. Therefore, new targeted therapies are urgently needed to cure these patients when they relapse. We focused our attention on cytokines of the CXC family, bearing a specific amino acid sequence of glutamic acid-leucine-arginine (ELR for short) immediately before the first cysteine of the CXC motif (ELR+). ELR+CXCL are small proteins that exert pro-angiogenic and pro-inflammatory effects via autocrine and paracrine circuits involving CXCR1/2 receptors [5,6]. These receptors are found on some tumour, endothelial, and immune cells. Furthermore, this signalling pathway induces phenomena such as proliferation, angiogenesis, inflammation, and immune tolerance. Therefore, targeting CXCR1/2 may be a promising strategy for cancer treatment. A novel pharmacological inhibitor (C29) of ELR+CXCL cytokine receptors, CXCR1/2, was shown to be non-toxic to normal cells. However, it efficiently causes cancer cell death in highly vascularised tumours (kidney and head and neck) [7]. Here, we propose to investigate the efficacy of this compound in different MB models.

### 1.2. Added Value of This Study

We have provided evidence that ELR+CXCL/CXCR1/2 levels correlate with shorter survival in MB patients. In addition, low concentrations of C29 efficiently affect the aggressiveness of MB cells from independent genetic subgroups that are either sensitive or resistant to radiotherapy. This compound reduces the growth of experimental MB in an ex vivo organotypic mouse model that includes the surrounding tissue and extracellular matrix involved in tumour progression. C29 is also able to efficiently cross the blood–brain barrier without accumulating in the brain. This new targeted therapy is leading the way for future clinical trials in relapsed patients, but also in all other patients in combination with conventional therapies to limit their toxic side effects.

### 1.3. Implications of All the Available Evidence

Considering the few molecular targets identified in MB patients and the fact that treatment options for relapsed patients are limited, our study proposes an alternative and innovative targeted therapy. This new inhibitor acts on cancer cell proliferation, migration, and invasion, but also on angiogenesis and immune tolerance, which are associated with metastatic spread and relapse. Given these facts, C29 could be moved into clinical trials to study its efficacy and toxicity in refractory patients. Ultimately, this drug could be offered to children as a first-line treatment in combination with current treatments or at relapse.

## 2. Introduction

Medulloblastoma (MB) is the most common malignant paediatric brain tumour. MB accounts for up to 25% of central nervous system (CNS) tumours in children and is rare in adults [8]. The cure rate is 70–80% when patients are diagnosed before metastatic spread, compared with 30–40% when MB is at an advanced stage [1,2,9].

Medulloblastomas are a heterogenous group of tumours comprised of four primary molecular subgroups: Wingless (WNT), Sonic Hedgehog (SHH), group 3, and group 4 [10]. WNT and SHH are each characterised by over-activation of the WNT and SHH signalling pathways, respectively [3]. MBs in groups 3 and 4 exhibit overexpression of N- and c-MYC, and chromosomal abnormalities [3,10,11,12,13], although no specific signalling pathway appears to be involved. Recently, a new molecular classification has been established, which provides for a subdivision of the individual molecular groups. The identification of 12 different molecular subtypes further complicates the stratification of MB in future clinical trials [14]. Among the genomic alterations detected in the different groups of MBs, inactivation of p53 is one of the most critical. P53 mutation is virtually absent from WNT MB [15]. This mutation appears in the three other groups and may account for a significant number of recurrent and fatal average-risk medulloblastomas [16].

Current treatment of MB combines surgical resection of the tumour with risk-adjusted craniospinal radiation (usually with photons and increasingly with protons) and chemotherapy. After surgery, irradiation, and adjuvant chemotherapy, patients with WNT MB have a 5-year progression free survival (PFS) of >95%, whereas the SHH and Group 4 patients have an intermediate PFS of 70–80%, as compared a 50–60% PFS for those with Group 3 MB [17,18]. The aggressiveness of the treatments leads to significant short- and long-term side effects such as neurocognitive and functional deficits [1,2]. Relapses are usually fatal, albeit very rare in the WNT MB subgroup [16].

Tumour angiogenesis correlates with metastasis and treatment resistance, resulting in shorter survival [19]. MBs are highly vascularised cancers that overexpress a variety of angiogenic factors, particularly VEGF, which play a key role in tumour growth [20]. Although in vitro results have been promising [21], anti-VEGF strategies have not improved MB outcome. Several factors may explain this ineffectiveness, in particular the high vascular permeability and poor perfusion of the MB vascular network [22]. By preventing the development of normal blood vessels, anti-angiogenic treatments are harmful to growing children. In addition, MBs express several pro-angiogenic redundant vascular endothelial growth factor factors (VEGF; VEGFB, VEGFC, angiopoietin 1 and 2, bFGF, etc.), which prevents the efficacy of anti-VEGF/VEGF receptor treatments as monotherapy [22]. Anti-angiogenic treatment, however, may be more effective and less harmful than anti-VEGF alone, as demonstrated by Levy and colleagues [23].

Inflammation and angiogenesis are closely linked in cancer [24]. The pro-inflammatory conditions in tumour tissue cause the expression of VEGF by tumour and microenvironment cells. The resulting hypervascularization promotes the transport of pro-inflammatory mediators into the tumour tissue. Therefore, inflammation promotes angiogenesis which, in turn, induces inflammation [25]. Our study focused on the ELR+CXCL/CXCR1/2 pathway. ELR+CXCL cytokines (CXCL1, 2, 3, 5, 6, 7, 8), which have the amino acid triplet glutamic acid-leucine-arginine (ELR) in their amino-terminal domain, are indeed involved in tumour growth, invasion, and metastasis formation [6]. These cytokines ensure their pro-angiogenic and pro-inflammatory effects by stimulating the G-protein-coupled receptors CXCR1 and CXCR2. The resulting autocrine and paracrine signals lead to activation of the ERK/MAP kinases and phosphoinositide 3-kinase (PI3K) signalling pathways, which promote tumour growth and inflammation [6], thus increasing the aggressiveness of various cancers [26].

CXCL8 (interleukin 8 or IL-8) is the founding member of the ELR+CXCL family. It promotes angiogenesis, tumourigenesis, and metastasis and is overexpressed in several tumours [27,28], including highly vascularised clear cell kidney carcinoma (ccRCC) [29]. It stimulates survival of prostate [30], ovarian [31], brain [32], and skin cancer cells [33]. CXCL1 increases proliferation of oesophageal [34], skin [35], bladder [36], stomach [37], and colorectal cancer cells [38]. CXCL7 promotes the progression of cholangiocarcinomas [39], lung [40], kidney [41], colon [42], and breast cancers [43].

CXCR1/2 are expressed by (a) cancer cells (cancer cell lines and cancer cells from tumour biopsies), which are critical for cancer progression and metastasis, (b) endothelial cells, to promote proliferation, survival and migration of vascular endothelial cells, leading to an angiogenic phenotype, (c) and inflammatory cells (neutrophils/tumour-associated macrophages), leading to an immune-tolerant microenvironment that promotes metastasis [44,45,46,47]. Therefore, these receptors have been described mainly for their role in angiogenesis and inflammation.

Our team developed a new pharmacological inhibitor of CXCR1/2. This inhibitor, designated C29, acts as an antagonist of ELR+CXCL binding to CXCR1/2 and was derived from a known inhibitor, SB225002, which acts on kidney cancer growth [7]. It was developed by a rational design based on scaffold hopping and docking studies on CXCR1 and CXCR2 structures. C29 has a triple effect on angiogenesis, inflammation, and tumour growth. It inhibits proliferation of numerous cancer cells and particularly cancer cells from highly vascularised cancers (kidney and head and neck) in vitro and in vivo [7,48].

The ELR+CXCL/CXCR1/2 axis has never been studied in MB, which are highly vascularised tumours. In the current study, we demonstrated that C29 has a significant inhibitory effect in several subtypes of MB in vitro and in vivo. Targeting this pathway may represent an alternative strategy for the first-line treatment of MB or after relapse by specifically targeting three cancer hallmarks: tumour cell proliferation, chronic inflammation, and angiogenesis [49].

## 3. Materials and Methods

### 3.1. Chemistry

C29 was synthesized at the Institute of Chemistry of Nice (ICN) following the described procedure and its analytical and spectrometric data were in accordance with previous characterisation data [7]. It is currently developed by the start-up Roca Therapeutics http://www.roca-therapeutics.org/ (accessed on 1 June 2019).

### 3.2. Cell Culture

#### Human Medulloblastoma Cells

Daoy (HTB-186) were purchased from American Tissue Culture Collection ATCC (Molsheim, France). ONS-76 cells were obtained from Dr. Celio Pouponnot (Curie Institute, Paris, France). Daoy and ONS-76 cells were cultivated in MEM (1X) + GlutaMAX medium (Invitrogen (Thermo Fisher, Waltham, MA, USA)), supplemented with 10% fetal bovine serum FBS (D. Dutscher, Brumath, France) and 1 mM sodium pyruvate (Gibco Life Technologies (Thermo Fisher)). D341Med cells (ATCC, HTB-187) were cultured in MEM (1X) + GlutaMAX (Invitrogen) medium supplemented with 10% FBS (D. Dutscher). HD-MB03 cells (DSMZ, Leibniz, Germany, ACC 740) were maintained in an RPMI medium + 1% Non-Essential Amino Acids NEAA (Thermo Fisher, Montigny-le-Bretonneux, France), supplemented with 10% FBS (D. Dutscher).

X-ray resistant Daoy and HD-MB03 cells were generated and cultured as described under the Appendix A and Methods section.

### 3.3. Flow Cytometry

#### CXCR1/2 Labelling

Cells were dissociated with 1X Accutase (HyClone HyQTase, Fisher Scientific (Thermo Fisher)). 10^7^ cells were resuspended in 500 µL of PBS, 0.5% BSA, and 2 mM EDTA. Cells were labelled with anti-human APC-conjugated anti-CXCR1 antibody, PE-conjugated anti-CXCR2 antibody, and the respective isotype control antibody (Miltenyi Biotec, North Rhine-Westphalia, Germany) for 10 min in the dark at 4 °C. Cells were washed twice and centrifuged at 300× *g* for 10 min. Cell pellets were resuspended in 300 µL of buffer (PBS, 0.5% BSA, 2 mM EDTA) for analysis by FACS Cytoflex (Beckman, Pasadena, CA, USA). The gating was realized with CytExpert 2.3 software associated with Cytoflex. The data were represented with the relative mean fluorescence intensity (MFI).

### 3.4. In Vitro Assays of Cell Behaviour

Several assays (proliferation, migration, invasion, XTT, pseudo-vessel formation, RT-qPCR, ELISA, Western blotting) were performed to analyse the MB cells behaviour and aggressiveness. The methodology of these different assays is described under the Appendix A and Methods section.

#### 3.4.1. Cerebellar Organotypic Model

To establish ex vivo cultures of cerebellar tissues, mouse cerebellums were surgically harvested and sectioned into 250 μm thick slices using a vibrating blade microtome (RRID:SCR_016495). To comply with the 3-R rule, we used supernumerary mice (aged 12 weeks) that were destined to be sacrificed. A spheroid formed from DsRed-expressing MB cells [50,51] was then grafted onto each cerebellum slice. These organotypic co-culture models were placed on inserts and maintained as previously described [51]. After daily exposure to C29 or DMSO for 5 consecutive days, tumour growth and invasion within the cerebellum slices were analysed over time using the JuLI™ Stage imaging system and the PHERAstar^®^ FS multiplate reader (λex 580 nm/λem 620 nm–fluorescence signal acquisition with a 15 × 15 matrix scanning mode).

#### 3.4.2. Quantification of C29 in Mouse Organs Postmortem

Calibration curve of C29

A 5 mM stock solution of C29 (1.743 mg/mL) in DMSO was prepared. This solution was diluted with a mixture of acetonitrile/water, 8/4, *v*/*v* to produce the solution with the highest concentration: 1250 ng/1.2 mL. Successive dilutions with acetonitrile/water, 8/4, *v*/*v* provided 5 more standards with concentrations between 50 ng/mL and 1250 ng/mL.

Preparation of C29 samples

Samples of cerebellum were prepared by dissolving minced biological material in 1.2 mL acetonitrile/water, 8/4, *v*/*v*. After sonication for 5 min, the vials were centrifuged (15,000 rpm) and the supernatant was filtered through a 4.5 Å PTFE microfilter.

Analysis of C29 samples

All standards and samples were analysed using a LC/MS system consisting of an Agilent 1100 HPLC chain with a degasser, a binary pump, a column oven, an automatic injector, and a UV detector. The column is a Phenomenex Gemini analytical column 3 µm C18 110 A 150 mm × 3 mm × 3 µm, at 25 °C. The solvents are water with 0.1% formic acid (solvent A) and acetonitrile with 0.1% formic acid (solvent B). The gradient is as follows: 10% B to 100% B over 8 min, 100% B over 6 min, 100% B to 10% B over 1 min, and 10% B over 3 min (total time: 18 min). The UV detection is carried out over the range of 190–400 nm (in 2 nm steps). The mass spectrometer is an LCQ Advantage (ThermoScientific, Waltham, MA, USA) of the “ion trap 3D” type, equipped with an electrospray source. The ionisation parameters are listed in Table 1:

Mass spectrometric detection was carried out in negative mode and in SRM (single reaction monitoring) mode according to the monitoring of the 380.9–194.2 transition. The isolation span of the ion was 2 mass units (193.2–195.2). The applied normalized collision energy was 30%. Each standard and sample were injected in duplicate. Each duplicate was separated by a blank and the concentration of the standards ranged from 0.034 ng/mL to 0.55 ng/mL. None of the blanks showed a signal above the detection limit. Signals were acquired based on their area and integrated using the ICIS algorithm according to the following parameters: smoothing points: 15; baseline window: 75 scans; area noise factor: 5; peak noise factor: 10. Calibration is external, and the regression was linear. The origin was included in the calibration.

#### 3.4.3. In Silico Analysis

CXCR1/2 and ELR+CXCL cytokines (CXCL1, 2, 3, 5, 6, 7, 8) relative mRNA expression levels were analysed using the R2: Genomics Analysis and Visualization Platform (http://r2.amc.nl (accessed on 1 June 2019)).

#### 3.4.4. Statistical Analysis

Statistical analyses were carried out using Prism 8 software. Results were reported as mean ± standard error (SEM) of at least three independent experiments. Unless otherwise stated, statistical analyses were performed using one- or two-tailed ANOVA tests with Dunnett’s multiple comparison test. Mann–Whitney analyses were performed for two independent groups. Results were considered significant if the *p*-value *p* < 0.05.

## 4. Results

### 4.1. Members of ELR+CXCL/CXCR Axis Are Overexpressed and Associated with Poor Prognosis in MB Patients

To gain insight into the importance of the ELR+CXCL-CXCR axis at MB, we first analysed the mRNA expression levels of several members of this pathway using several public databases generated from patients. Our analysis was devoted to both receptors (CXCR1/2) and ELR+CXCL cytokines (CXCL1, 2, 3, 5, 6, 7, 8) selected from six independent datasets (M1–M6) and compared with datasets from a normal brain (NB) and a tumour brain (TB). Our screening showed that expression of CXCR1/2 and ELR+CXCL cytokines was higher in TB than in NB (Figure 1A,B). Next, we restricted our analysis to compare MB patients (M1–M6) with NB. CXCR1 and CXCL7 expression was higher in all datasets compared to NB (*p* < 0.0001) and CXCL8 expression was higher in four out of six datasets (*p* < 0.0001). Surprisingly, CXCL1, 2, 3, and 5 expressions were increased or decreased depending on the dataset (Figure 1A,B).

To determine overall survival (OS) as a function of mRNA expression, we studied 763 patients from the Cavalli database [14]. We found that high CXCR2 mRNA levels correlated with shorter median OS, (115 months in the high expression patients compared to 264 months in the low expression patients, *p* = 0.01 (Figure 1C)). For CXCR1, we observed a similar trend (low expression 264 months vs. high expression 180 months) but this did not reach statistical significance (Figure 1C). With the exception of CXCL2, low expression of ELR+CXCL tended to result in a better median OS (Appendix A, CXCL1: 175 (low) vs. 136 (high) months; CXCL2; 168 (low) vs. not reached (NR, high) months; CXCL3; 180 (low) vs. 167 (high) months; CXCL5; NR (low) vs. 168 (high) months: CXCL6; 180 (low) vs. 167 (high) months: CXCL7; 175 (low) vs. 160 (high) months: CXCL8; and 175 (low) vs. 150 (high) months).

To test whether OS was related to a specific molecular subgroup, we assessed the prognosis of MB patients by the expression of mRNAs from the ELR+CXCL-CXCR1/2 pathway.

For SHH and group 4 patients, overexpression of CXCL3 and CXCL1 correlated with a shorter OS (*p* = 0.022 and *p* = 0.0072, respectively). In contrast, in group 3, overexpression of CXCL7 correlated with a longer OS (*p* = 0.045). To further our analysis, we analysed the differences in the percentage of OS five years after diagnosis as this time point is crucial for patient outcome (Table 2). We also used this method to avoid bias, as the median OS for SHH patients was not reached. A positive number (+1) was arbitrarily assigned if overexpression correlated with at least a 10% increase in survival at 5 years. A negative number (−1) was assigned if overexpression correlated with at least a 10% decrease in survival after five years. Percentages below this threshold were considered zero. This value was negative in the combined subgroups (value = −2), in the SHH subgroup (value = −5), and in group 4 (value = −5). It was zero in the WNT subgroup and group 3. Using this criterion, the ELR+CXCL-CXCR1/2 pathway did not appear to play a key role in the WNT subgroup and in group 3 (Appendix A). Although this scoring method still needs to be validated, it provided some clues to the importance of the ELR+CXCL-CXCR1/2 pathway in MB patients.

Overall, our results showed that several members of the ELR+CXCL-CXCR1/2 axis are overexpressed in MB patients compared with a healthy brain. Some cytokines are associated with poor prognosis depending on the genetic subgroup, whereas some prove beneficial. These findings suggest that targeting the ELR+CXCL-CXCR1/2 pathway may be a promising therapeutic strategy, but only for patients of certain genetic subgroups.

### 4.2. Expression of ELR+CXCL-CXCR1/2 Family Members in MB Cells

To further investigate the importance of the ELR+CXCL-CXCR1/2 axis in MB, we resorted to cell lines because of the rarity and the difficulty of directly accessing MB patient samples. We used four different cell lines, Daoy, ONS-76 (two independent cell lines of the subgroup SHH), HD-MB03, and D341Med (two independent cell lines of group 3), to account for variability due to tumour heterogeneity.

Since CXCR1 and 2 are active when expressed on the cell membrane, their presence was quantified by FACS analyses and presented as relative mean fluorescence intensity (MFI, arbitrary unit; Figure 2A). CXCR1 was detected only in D341Med cells and was absent in all other cell lines. In contrast, the CXCR2 receptor was expressed in all cell lines tested (Figure 2A).

The qPCR analysis evaluating CXCR1 and 2 mRNA levels showed a similar trend for all MB cells, and confirmed the previous FACS results (Figure 2B).

The levels of CXCR1 and 2 (R1, 2) and of ELR+CXCL cytokines (L1, 2, 3, 5, 7, 8; L6 was undetectable) were evaluated by qPCR and showed quite heterogenous expression between the different MB cell lines (Figure 2B). Overall, Daoy and ONS-76 expressed more ELR+CXCL cytokines than HD-MB03 and D341Med cells.

To confirm our results, we measured the secretion of ELR+CXCL cytokines by ELISA (Figure 2C). Daoy secreted significantly more CXCL1 and CXCL2 (*p* < 0.01) than any other MB cell line. Daoy and ONS-76 secreted significantly more CXCL5 and CXCL8 than the other cell lines (*p* < 0.001).

The secretion of CXCL7 was comparable among the four cell lines, but it was very low compared to the other cytokines (below 100 pg/mL/10^6^ cells). It is possible that CXCL7 is not the major player in Daoy and ONS-76 cells, whereas it might predominate in HD-MB03 and D341Med cells. Daoy and ONS-76 cells secreted significantly more CXCL8 than HD-MB03 and D341Med cells, *p* < 0.001. CXCL8 is the most prevalent cytokine secreted by these cell lines. These results clearly show that the SHH cell lines express more ELR+CXCL cytokines than cells in Group 3.

Our screening showed that ELR+CXCL-CXCR1/2 signalling, as in patients, is generally induced in a heterogeneous manner. However, the CXCL8-CXCR2 axis was found to be overrepresented in SHH cell lines, suggesting that targeting this signalling may be a relevant therapeutic strategy, at least for this molecular subgroup.

### 4.3. Inhibition of Cell Metabolism by the Novel Pharmacological CXCR1/2 Inhibitor C29 in MB Cells

A novel pharmacological competitive inhibitor of CXCR1/2, 1-(3-chlorophenyl)-3-(6-nitrobenzo[d]thiazol-2-yl) urea, designated C29, was recently described by our team as effective in renal and head and neck tumours [7]. This diarylurea derivative is the lead compound of a large series of analogues, developed by scaffold hopping and docking experiments using CXCR1 and CXCR2 receptor structures. As described previously, C29 was more efficient than other commercially available CXCR1/2 inhibitors in various tumour cells [7].

To evaluate the efficacy of this compound in MB cells, we first determined the relative IC_50_ values for each cell line using XTT assays (Figure 3A–E). The XTT assay is used to measure cellular metabolic activity (overall activity of mitochondrial dehydrogenases) as an indicator of cell viability, proliferation, and cytotoxicity. C29 appears to be less effective in ONS-76 than in all other cell lines at 24 and 48 h (IC50 = 17.9 and 6.8 µM, respectively), whereas HD-MB03 cells appear to be more sensitive to this inhibitor than other cells at 24 and 48 h (IC50 = 2.6 and 1.8 µM, respectively). After 72 h, C29 appears to be more effective on Daoy cells (IC50 = 0.9 µM) and less effective on D341Med cells (IC50 = 4.6 µM). Unexpectedly, C29 stimulated metabolism in D341Med cells over a longer period of time (72 h), but only at low concentrations (0.1 to 2.5 µM). At higher concentrations, it consistently inhibited cell metabolism as in the other cell lines. Overall, IC50 values for all cell lines were in the micromolar range, making C29 a very efficient agent to inhibit cell metabolism and viability.

We then determined the selectivity index (SI), which is the IC50 value for normal cells divided by the IC50 value for tumour cells [52], to assess the overall toxicity on normal tissues. These IC50 values determined by XTT assays or measurement of cell death by PI staining were nearly equivalent (Figure 3E,F). Primary mouse astrocytes (C8D1A) were selected based on their nervous system lineage. The IC50 of C29 for these cells was 19.32 µM at 48 h of treatment. The SI was thus 6.44 for Daoy, 2.84 for ONS-76, 10.7 for HD-MB03, and 3.94 for D341Med, indicating selectivity towards tumour cell death at the doses considered. These values are also consistent with those determined for renal cancer cells [7].

### 4.4. C29 Reduces Proliferation, Migration, and Invasion of Naïve MB Cells

To confirm the efficacy of C29 on various parameters of cancer cell aggressiveness, we first performed cell proliferation assays at longer time points. We found that pre-incubation of 1 µM C29 prior to cell seeding completely inhibited proliferation of all tested cell lines (Figure 4A).

At the same concentration, C29 also inhibited clone formation in Daoy (*p* < 0.001) and HD-MB03 cells (*p* < 0.001), whereas in ONS-76 cells, some clones were maintained and disappeared completely at 2.5 µM (*p* < 0.001) (Figure 4B). At a concentration of 1 µM, C29 blocked CXCL8-induced migration in both Daoy and ONS-76 cells (Figure 4C, *p* < 0.001). Since HD-MB03 cells could not migrate (Appendix A), C29 was not tested in these cells and their resistant derivatives. Since the ability to migrate is a prerequisite for invasion during the metastatic process, we next examined a spheroid invasion model. We found that C29 affected the invasion ability of Daoy cells from day 2 and ONS-76 cells from day 7 (Figure 4D).

HD-MB03 cells formed spheroids but were unable to invade the Matrigel matrix (Appendix A). D341Med cells did not form spheroids as they grow in suspension.

Next, we investigated the signalling pathways affected by C29 to further characterise its mechanism of action. The AKT and ERK signalling pathways are mainly activated by CXCR1/2 stimulation. We showed that C29 inhibited ERK phosphorylation in Daoy and HD-MB03 cells (Figure 5A,B) and ERK and AKT phosphorylation in ONS-76 cells, indicating their activity (Figure 5C). These results are consistent with the inhibition of proliferation and induction of cell death described in Figure 3 and Figure 4. They highlight the importance of these two signalling pathways for MB cell survival and proliferation.

Hence, C29 appears to have a strong effect on naïve MB cells via an autocrine or paracrine regulatory circuit dependent on the ELR+CXCL-CXCR1/2 axis.

### 4.5. C29 Inhibits the Formation of Pseudo-Vascular Structures

The concept of vasculogenic mimicry corresponds to vessel-like formation by endothelial cells or some tumour cells that exhibit invasive capacity [53,54]. Daoy, ONS-76, and HD-MB03 cells were able to organize as pseudo-vessels [55], and D341Med cells were unable to do so.

The efficacy of C29 was also assessed by the ability of the MB cell lines to form a pseudo-vasculature (Figure 6). It inhibited this ability of Daoy cells at 1 µM and especially at 2.5 µM (*p* < 0.001). In ONS-76 cells, some vascular structures were retained at 1 µM, whereas the cells completely lost this ability at 2.5 µM (*p* < 0.001). C29 also significantly reduced the number of meshes formed by HD-MB03 cells at 1 µM (*p* < 0.001).

These results suggest that C29 inhibits the ability of tumour cells to organize into hybrid vascular structures, an independent pathway involved in the metastatic spread of tumour cells.

### 4.6. C29 Is Active in the Ex Vivo Organotypic Model of the Cerebellum and Crosses the Blood–Brain Barrier In Vivo

To investigate the activity of C29 under more clinically relevant conditions, we developed an organotypic cerebellum model in which MB spheroids stably expressing DsRed were grafted into slices of healthy mouse cerebellum (Figure 7A). These ex-vivo cultures were treated with 5 µM C29 daily for five consecutive days. C29 reduced the growth of MB tumour as well as the invasion of MB cells into the cerebellar tissue over time. After fourteen days, tumour expansion of ONS-76 and HD-MB03 was reduced by 28.9 ± 4.5% and 34.4 ± 8.8%, respectively (*p* < 0.05; Figure 7B–E), confirming the efficacy of C29 in inhibiting tumour progression.

A critical and limiting feature of a drug targeting MB is its ability to cross the blood–brain barrier (BBB) and reach the tumour in the brain. We demonstrated by direct post-mortem dosing of mice cerebellum that C29 was present in significant amounts 3 h after oral administration of 200 mg/kg (Table 3). However, after 5 days of treatment, 24 h after the last administration, no more C29 was found in the cerebellum of the mice. These results strongly suggest that C29 can cross the blood–brain barrier, probably by passive diffusion given its lipophilicity (log D ≈ 3), but that it does not accumulate upon repeated administration. These results are crucial for the next steps in drug development, as they allow for systemic administration of C29, which is strongly preferred to topical administration for a molecule targeting MB.

### 4.7. C29 Is Effective in Radiation-Resistant MB Cell Lines

Relapse after radiation is a major problem. Therefore, any treatment that can induce the death of tumour cells that have survived multiple cycles of radiation is an important breakthrough. To test the relevance of C29 after relapse, we generated radiation-resistant Daoy and HD-MB03 cells to simulate relapse after radiotherapy [55]. Two independent populations were generated for each cell line. These resistant cell lines (DR1, DR2, HR1, HR2) were found viable and resistant after more than 10 cycles of photon irradiation at 8 Gy [55]. ELR+CXCL-CXCR1/2 expression was measured in these cell models. The production of CXCL1, 2, 5, 7, and 8 was not altered compared to the corresponding naïve cell lines (Daoy and HD-MB03, Appendix A). The expression of CXCL1, 2, 5, and 8 was as high in DR1 and DR2 as in Daoy and as low in HR1 and 2 as in HD-MB03. CXCL7 secretion was low in all naïve and radiation-resistant cells (Daoy: 50.9 ± 5.4 pg/mL/10^6^ cells, DR1: 57.9 ± 20 pg/mL/10^6^ cells, DR2: 56.4 ± 25.1 pg/mL/10^6^ cells, HD -MB03: 51 ± 15 pg/mL/10^6^ cells, HR1: 44.5 ± 18.7 pg/mL/10^6^ cells, and HR2: 38.3 ± 10.5 pg/mL/10^6^ cells, ns).

As with naïve cells, we determined the IC50 for DR1, DR2, HR2, and HR2 and their SI compared to astrocytes. According to their specific IC50 (Figure 8A), the SI were 3.9 for DR1 and DR2, 7.7 for HR1 and 2.8 for HR2, respectively.

Subsequently, we showed that C29 can reduce proliferation of radiation-resistant cells as efficiently as on naïve cells (Figure 8B). C29 also inhibited the ability of these resistant cells to form clones (Figure 8C). In radiation resistant Daoy cells (DR1, DR2), C29 was effective at 1 µM (*p* < 0.001) and completely prevented clone formation at 2.5 µM (*p* < 0.001). In HR1, C29 reduced the number of clones at 1 µM (*p* < 0.001) and extinguished them at 2.5 µM (*p* < 0.001). In HR2, C29 significantly reduced clone formation only at 2.5 µM (*p* < 0.001).

We observed the same trend in transmigration (Figure 8D). At 1 µM, CXCL8-dependent migration was significantly inhibited by C29 (*p* < 0.01), and even more significantly at 2.5 µM (*p* < 0.001).

According to these and previous results, C29 appears to be effective in naïve and radiation-resistant MB cells.

## 5. Discussion

Despite the improvement of various therapies, the management of MB relapse remains delicate, and treatment options are limited. Innovative therapies can significantly improve the outcome for patients who are at a therapeutic impasse due to the lack of targeted therapies for relapse. Based on our findings, we believe that targeting the ELR+CXCL/CXCR1/2 pathway is an interesting therapeutic option. We report here that a new CXCR1/2 inhibitor called C29, developed in our laboratory [56], significantly reduced proliferation, migration, invasion, and formation of pseudo-vessels in MB cells in vitro. C29 also reduced the growth and invasion of experimental tumours in an ex vivo cerebellar organotypic model of MB. These observations and the high expression of the CXCR2 receptor in MB suggest that C29 may represent a valid therapeutic option against MB.

To support our conclusions regarding the inhibitory effect of C29 on CXCR2, we performed experiments to knock-down this receptor. Despite several attempts with different protocols and several commercial and validated siRNAs, we had to reluctantly discontinue these experiments because, surprisingly, none of the siRNAs that we tried decreased the expression of the CXCR2 receptor. Since the expression of CXCR1 was so low in our MB cells, we assumed that this receptor may not play an important role and did not try to knock it down. However, it is important to note that C29 had no effect on the proliferation and survival of uveal melanoma cells (Mel202), which display low expression of CXCR1/2 [56]. This suggests that CXCR1 and 2 are the only targets of C29 and suggests that the efficacy of C29 on MB is related to the presence of the CXCR2 receptor. However, to further confirm this claim, it is very important to develop MB cell models that lack CXCR2 expression. Our efforts are currently directed towards developing such models (CRISPR-Cas9 experiments), and we will report the results in due course.

We showed that CXCR2 is expressed in all cells of the tumour niche, including of course tumour cells as previously described [7,56] and highlighted in this manuscript, but also endothelial and inflammatory cells and tumour-associated fibroblasts. Such expression leads to multiple interactions in the tumour microenvironment, causing deleterious shaping of tumour niche cells, but also pro-tumour [57] and immunotolerant secretome. In addition, CXCR1/2 triggers stem cell formation, as previously described in aggressive breast cancer [58]. Thus, by targeting CXCR2, we can simultaneously inhibit multiple features of cancers, including stem cell/proliferation/invasion/migration of tumour cells, angiogenesis, fibrosis, a feature of cancers with poor prognosis [59], and immune tolerance. Thus, ELR+CXCL cytokines and receptors are deleterious in several cancers, particularly metastatic renal and head and neck tumours [7].

This general scheme does not seem to be so simple. According to an analysis of databases available online, ELR+CXCL/CXCR1/2 may be beneficial or detrimental depending on the specific genetic subgroup of MB. Counterintuitively, it is not really involved in the most aggressive Group 3. MB are sparsely infiltrated by immune cells and are known as “cold tumours”. In addition, the blood–brain barrier is a natural blockade that prevents immune cells from infiltrating and radio/chemotherapy treatments destroy the immune system. Furthermore, corticosteroid therapy, which is often used to prevent headaches and vomiting, reinforces the immunosuppressive context. The role of myeloid cells is controversial, as they have been described as pro- or antitumour depending on the genetic subgroup [60]. The main role of ELR+CXCL is to attract neutrophils and promote positive inflammation. However, the generation of M2 macrophages or regulatory T lymphocytes, which is a consequence of maintaining a specific secretome, is detrimental in several tumours, including MB. We believe that a beneficial effect depends on a specific design of the tumour microenvironment that drives immune cells into the core of the tumour by increasing blood vessel density, allowing non-exhausted T cells to exert their anti-tumour effect.

A very important point for translating our research into the clinic was the ability of C29 to cross the blood–brain barrier. This property can generally be a double-edged sword, as accumulation of the drug can lead to toxicity. Our result clearly showed that a dose of 200 mg/kg via oral gavage resulted in C29 being detected in the brain, implying that the parameters of solubility, absorption, distribution, and metabolism allowed this to occur. This result suggests that oral administration is possible, a very important parameter for patient comfort that does not require hospitalization. It would be important to develop a specific formulation to increase the bioavailability of the drug and achieve a higher concentration in the cerebellum for maximum therapeutic effect.

Furthermore, C29 cannot be detected in the cerebellar extract after chronic exposure to the drug for five days, 24 h after the last administration of the drug. This result supports the assumption of a safe C29 therapeutic efficacy, without potentially toxic accumulation in the brain.

Although the use of different cell lines allowed us to establish a proof of concept and get as close as possible to the genetic heterogeneity of the pathology, the exploitation of an innovative ex vivo model made it possible to add a level complexity to the study. This model of MB spheroid grafting onto mouse cerebellar sections includes the original surrounding tissue and extracellular matrix that participate in tumorigenesis. The study of the efficacy of C29 on this model, combined with BBB crossing experiments, does not replace the complexity of a whole organism, but constitutes solid preclinical validations to predict the efficacy of a compound in vivo. The primary treatment for MB is surgical resection. Thus, the development of innovative treatments must aim to treat metastases and any residual tumour. To mimic a metastasis model, subcutaneous xenograft models are an easy model to establish the efficacy of C29 in vivo. However, in view of the many interactions between the lymphatic, blood, and immune systems, extending this work to orthotopic xenograft into the cerebellum of immunocompetent models will allow for an integrated study of these systems.

Limiting the toxic effects of a given treatment is a real concern as patients with MB suffer the side effects of multimodal therapies including surgery and chemo/radiotherapy. De-escalating dosages should be tested on MB to reduce the consequences of such aggressive treatments. Since C29 is equally effective in naïve and radiation-resistant cells, we suspect that it might act as a radio-sensitiser of tumour cells. Provided it was the case, which is left to demonstrate in upcoming in vivo experiments, C29 could be used to potentialize the action of radiations. According to R. Jain’s theory, anti-angiogenic drugs do not destroy the vascular network but normalize it, resulting in better blood flow to the tumour [61]. Therefore, conventional chemotherapies used for cancers of different origins had better access to the cells in the core of the tumour. C29 should normalize the abnormal vascular network and have a better direct effect on the tumour cells. Better oxygenation of the tumour is also favourable for better efficacy of radiotherapy.

We have recently observed that stem cells are a deleterious feature of MB and the transcription factor OCT4 plays a key role in this mechanism [62]. Among the various genes induced in stem cells under specific culture conditions, some ELR+CXCL cytokines such as CXCL1 were upregulated. This finding suggests that the ELR+CXCL-CXCR1/2 pathway may be involved in the stemness of MB and consequently in their relative aggressiveness, which is consistent with corresponding findings on breast cancer stem cells [58].

This specific mechanism reinforces our assumption that the therapeutic index of C29 is multifaceted, mainly by inhibiting stem cell formation.

In conclusion, our manuscript proposes an alternative therapy for MB patients after relapse to current treatments. We thus describe another indication for the use of C29, as we have demonstrated its efficacy in renal and head and neck tumours. Further experiments are needed before administration to children, but our experiments pave the way for future clinical trials, first in refractory patients, and, afterwards, in combination with conventional treatments to limit their potential toxic side effects.

## Figures and Tables

**Figure 1 cells-11-03933-f001:**
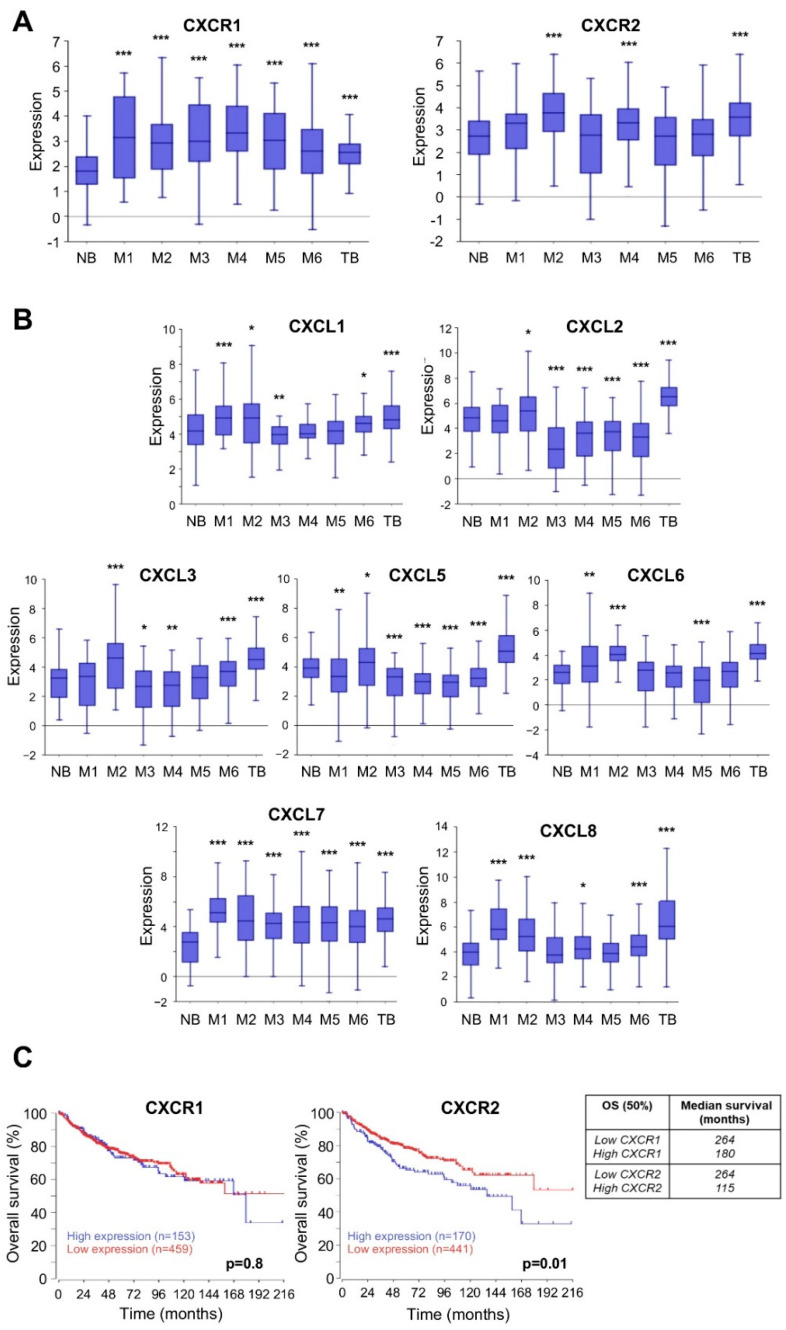
Members of ELR+CXCL/CXCR axis are overexpressed and associated with poor prognosis in MB patients. (**A**,**B**). Box-dot plot showing mRNA expression level of CXCR1, CXCR2, CXCL1, CXCL2, CXCL3, CXCL5, CXCL6, CXCL7, and CXCL8 (R2 platform) in healthy brain (Normal brain, NB) (n = 172 retrieved from Berchtold database), MB patients (M1–M6), and tumour brain (TB) (n = 550 retrieved from Madhavan database). M1: (n = 31) from Hsieh database, M2: (n = 51) from den Boer database, M3: (n = 57) from Delattre database, M4: (n = 62) from Kool database, M5: (n = 76) from Gilbertson database and M6: (n = 223) from Pfister database. *p* < 0.001. One-way analysis of variance (ANOVA). Data are means ± SD. (**C**). Analysis of overall patient survival (Kaplan-Meier curves) as a function of the rate of CXCR1 or 2 mRNA. Data are from the R2: Genomics Analysis and Visualization Platform (http://r2.amc.nl (accessed on 1 June 2019)), retrieved from the Cavalli database and analysed as combined subgroups, cut off = last quartile. Data were analysed by two-way ANOVA. *: *p* < 0.05; **: *p* < 0.01; ***: *p* < 0.001.

**Figure 2 cells-11-03933-f002:**
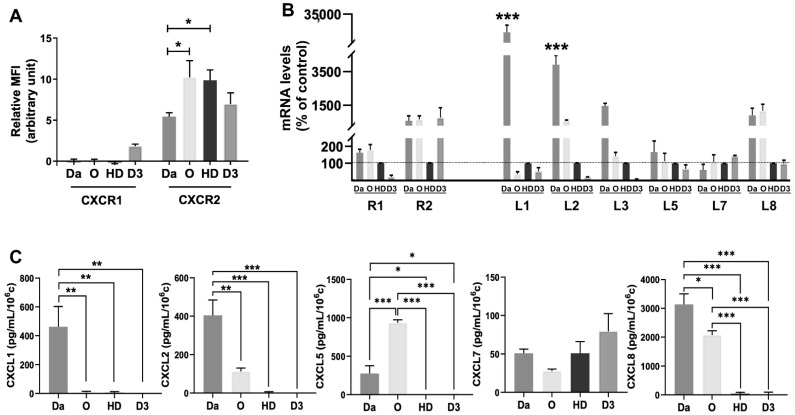
MB sensitive and irradiation-resistant cells expressed ELR+CXCL-CXCR markers. (**A**). Labelling of CXCR1/2 membrane receptors by flow cytometry on Daoy (Da, n = 4), ONS-76 (O, n = 4), HD-MB03 (HD, n = 3), and D341Med (D3, n = 3) cells. (**B**). Human CXCR1/2 (R1/R2) and ELR+CXCL (L1, L2, L3, L5, L7 and L8) mRNA expression level in Daoy, ONS-76, HD-MB03, and D341Med cells. The mRNA level was determined by qPCR. Results are expressed as percentage of control (HD-MB03), n = 3. (**C**). ELR+CXCL (CXCL1, 2, 5, 7, 8) dosage by ELISA assay in Daoy, ONS-76, HD-MB03, and D341Med WT cells, n = 3. Statistics were performed by two-way method ANOVA. * *p* < 0.05, ** *p* < 0.01, *** *p* < 0.001. For this figure, Daoy = Da, ONS-76 = O, HD-MB03 = HD, D341 = D3.

**Figure 3 cells-11-03933-f003:**
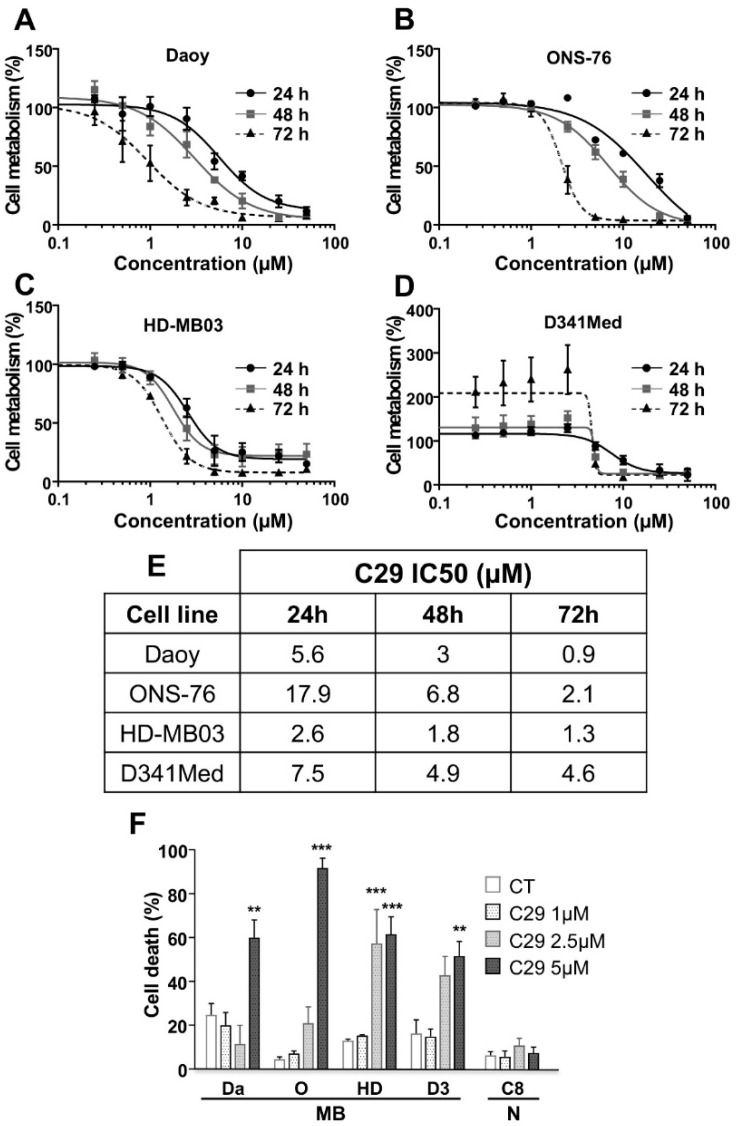
A new pharmacological inhibitor of CXCR1/2, C29, inhibits the metabolism of MB cells. (**A**–**D**). Cytotoxic effect induced by C29 on Daoy (**A**), ONS-76 (**B**), HD-MB03 (**C**), and D341Med (**D**) cells. Cells were treated with increasing doses of the inhibitor for 24, 48, and 72 h. Cytotoxicity was measured using the XTT assay. Absorbance measurements were performed at 450 nm, n = 3. (**E**) Table with the different IC50 values for all conditions. (**F**) Cell death after 48 h of treatment with control (DMSO, CT) or C29 1, 2.5 or 5 µM treatment, on MB cells, and normal cells (N), n = 3. Statistics were performed using the one-way method ANOVA. ** *p* < 0.01, *** *p* < 0.001. For this figure, Daoy = Da, ONS-76 = O, HD-MB03 = HD, D341 = D3, C8D1A = C8.

**Figure 4 cells-11-03933-f004:**
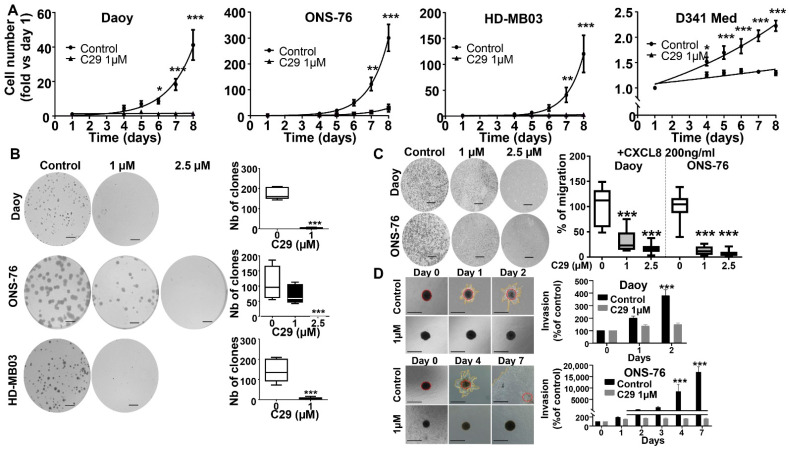
C29 reduces proliferation, migration, and invasion of naïve MB cells. (**A**) Proliferation of Daoy, ONS-76, HD-MB03, and D341Med wild-type cells after treatment with 1 μM C29 or the control (DMSO), n = 3. (**B**) Clonogenic assay on Daoy (n = 4), ONS-76 (n = 3) and HD -MB03 (n = 5) cells after treatment with control (DMSO) or C29 (1 μM or 2.5 μM). Scale = 500 μm. (**C**) Cell migration through the Boyden chamber of Daoy and ONS-76 cells specifically attracted by 200 ng/mL CXCL8. Results are expressed as percentage of control (DMSO) (n = 3). Scale = 1000 μm. (**D**) Spheroid invasion using 3D culture cell assays on Daoy (n = 4) and ONS-76 cells (n = 3) treated with control (DMSO) or C29 at a concentration of 1 μM on day 0, 1, and 2. Results are expressed as percentage of day 0, n = 4. Scale = 500 μm. Statistics were performed using Student *t*-test for 2 comparisons only or ANOVA test for multiple comparisons. * *p* < 0.05, ** *p* < 0.01, *** *p* < 0.001.

**Figure 5 cells-11-03933-f005:**
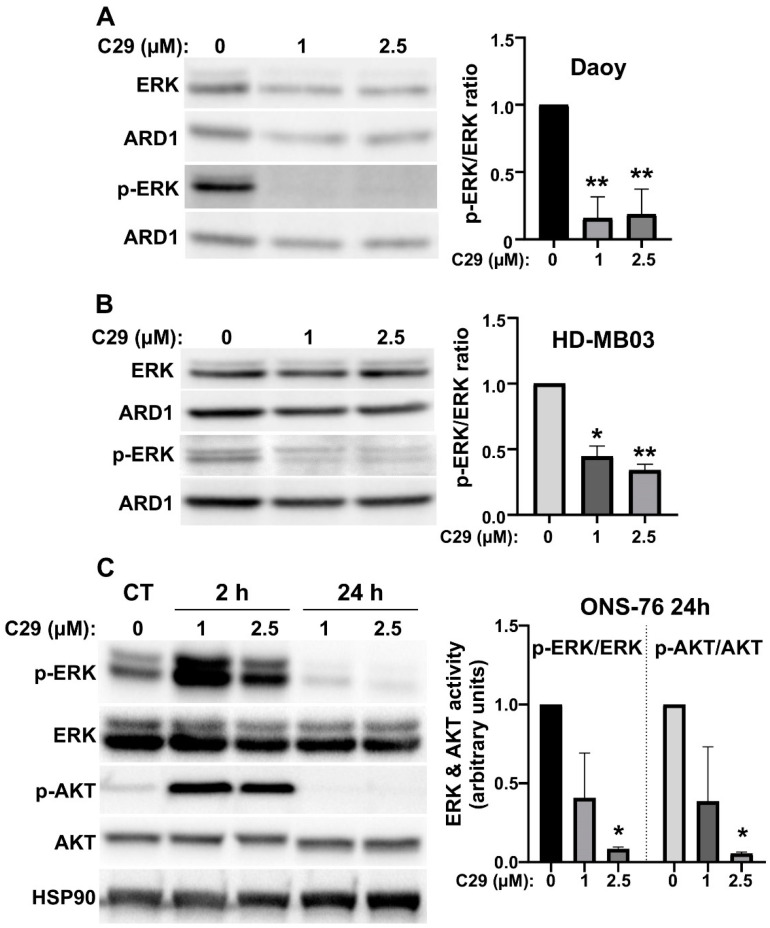
C29 inhibits signalling pathway that stimulates in cell proliferation and survival. (**A**–**C**). Daoy (**A**), HD-MB03 (**B**), and ONS-76 (**C**) cells were treated with C29 (1 or 2.5 μM) for 2 h (**A**–**C**) or 24 h (**C**), n = 3. The expression of p- ERK and p-AKT, relative to ERK and AKT, (respectively) was used to describe the relative activation of the pathways. ARD1 and HSP90 were used as loading controls. Statistics were performed using Student *t*-test for two comparisons only or ANOVA test for multiple comparisons. * *p* < 0.05, ** *p* < 0.01.

**Figure 6 cells-11-03933-f006:**
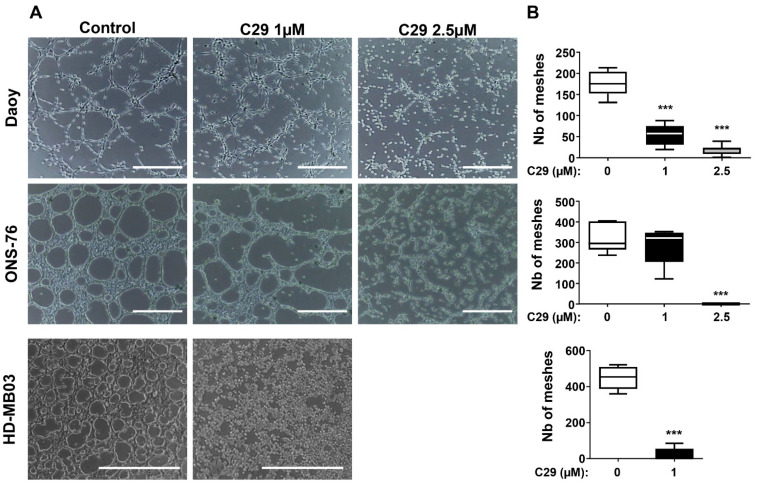
C29 inhibits the formation of pseudo vessel structures. (**A**) Formation of pseudo-vessels by Daoy cells, ONS-76, and HD-MB03 cultured on Matrigel and treated with Control (DMSO) or C29 (1 or 2.5 µM); n = 3. (**B**) Number of meshes (geometric units of the net formed by the cells on the matrigel surface) were counted with ImageJ. Statistics were performed using Student *t*-test for only 2 comparisons or ANOVA test for multiple comparisons. *** *p* < 0.001, scale = 500 μm.

**Figure 7 cells-11-03933-f007:**
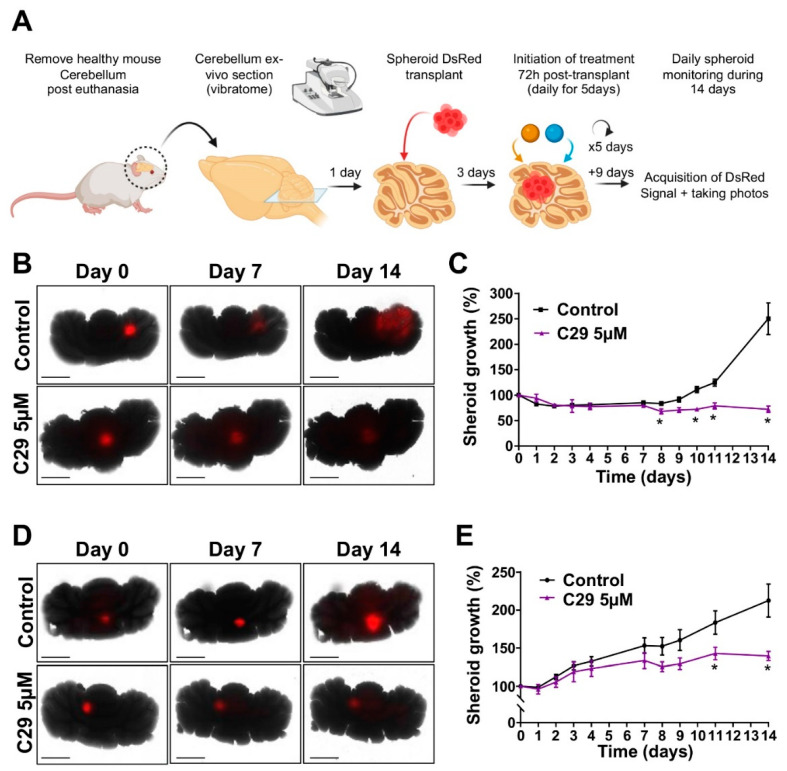
C29 is effective in ex vivo cerebellar organotypic model. (**A**) Experimental setup. (**B**,**C**) Growth of 3D ONS-76 spheroids expressing DsRed transplanted onto mouse cerebellar slices. Then, 72 h after transplantation, spheroids were treated with 5 µM C29 or control (DMSO) for 5 days to assess the growth and dissemination of spheroids. (**B**) Spheroid growth was monitored by measuring DsRed fluorescence (PheraStar) over 14 days (**C**) Statistics were performed using Student *t*-test. * *p* < 0.05, n = 5. (**D**,**E**) Growth of 3D HD-MB03 spheroids expressing DsRed transplanted onto mouse cerebellar slices. Then, 72 h after transplantation, spheroids were treated with C29 (5 and 10 µM) or control (DMSO) for 5 days. Daily (14 days) images, 30 in total, of spheroid growth in each well (JULISdays) were taken (**D**) Spheroid growth was monitored by measuring DsRed fluorescence (PheraStar) for 14 days (**E**) Scale bar = 250 µM. Statistics were performed using Student *t*-test. * *p* < 0.05, n = 5.

**Figure 8 cells-11-03933-f008:**
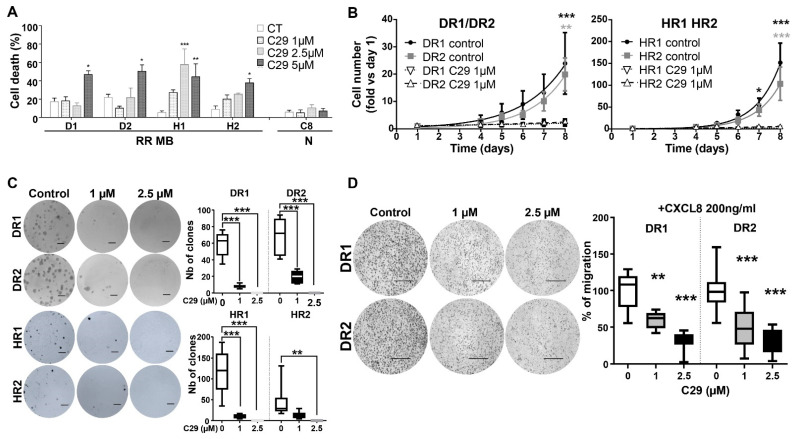
C29 is efficient on radioresistant MB cell lines. (**A**) Cell death after 48 h of treatment with control (DMSO, CT) or C29 1, 2.5, or 5 µM on radiation-resistant cells (RR) and normal cells (N), n = 3. Statistics were performed using the one-way method ANOVA. * *p* < 0.05, ** *p* < 0.01, *** *p* < 0.001. For this figure, DR1 = D1, DR2 = D2, HR1 = H1, HR2 = H2, and C8D1A = C8. (**B**) Proliferation of DR1/DR2 (**A**) and HR1/HR2 (**B**) radio-resistant cells after 1 µM C29 treatment or control (DMSO), n = 3. (**C**) Clonogenic assay on DR1, DR2 (n = 3) and HR1, HR2 (n = 3) cells after treatment with control (DMSO) or C29 (1 µM or 2.5 µM). (**D**) Cell migration through the Boyden chamber of DR1 and DR2 cells specifically attracted by 200 ng/mL CXCL8. Cells were treated with control (DMSO) or C29 (1 and 2.5 µM). Results are expressed as percentage of control (n = 3). Scale = 1000 μm. Statistics were performed using Student *t*-test for 2 comparisons only or ANOVA test for multiple comparisons. * *p* < 0.05, ** *p* < 0.01, *** *p* < 0.001.

**Table 1 cells-11-03933-t001:** Ionisation parameters used for mass spectrometry detection of C29.

Capillary Temp (°C)	275
Sheath Gas Flow (u.a.):	50
Aux/Sweep Gas Flow (u.a.)	10
Negative Mode	
Source Voltage (kV)	4.50
Source Current (uA)	80.00
Capillary Voltage (V)	−10.00
Tube Lens Offset (V)	−50.00

**Table 2 cells-11-03933-t002:** ELR+CXCL/CXCR expression in patients in relation to prognosis. Summary table showing 5-year overall survival (OS) as a function of ELR+CXCL/CXCR mRNA expression in the different subgroups of MB patients. The last lane is an arbitrary grading of the prognosis of the MB patients according to their CXCL and CXCR expression (positive trend = +1, negative trend = −1; cases coloured in black, if the difference between OS is less than 10% = 0; cases coloured in grey). A negative score corresponds to a poor prognosis, a positive score is associated with a good prognosis. This analysis was done with the software R2 in the Cavalli database, cut off = last quartile.

	All Groups (n = 612)			WNT (n = 63)			SHH (n = 172)			Group 4 (n = 264)			Group 3 (n = 113)		
Last Quartile		OS (%) 5y	*p*-Value		OS (%) 5y	*p*-Value		OS (%) 5y	*p*-Value		OS (%) 5y	*p*-Value		OS (%) 5y	*p*-Value
CXCR1	Low	76%	0.799	Low	98%	0.917	Low	80%	0.904	Low	78%	0.159	Low	55%	0.294
	High	72%		High	100%		High	80%		High	68%		High	68%	
CXCR2	Low	79%	0.001	Low	98%	0.724	Low	79%	0.203	Low	79%	0.157	Low	55%	0.583
	High	68%		High	100%		High	82%		High	65%		High	60%	
CXCL1	Low	78%	0.283	Low	98%	0.153	Low	84%	0.93	Low	82%	0.0072	Low	54%	0.528
	High	70%		High	100%		High	74%		High	58%		High	67%	
CXCL2	Low	78%	0.305	Low	96%	0.148	Low	84%	0.168	Low	76%	0.854	Low	58%	0.582
	High	69%		High	100%		High	72%		High	76%		High	52%	
CXCL3	Low	78%	0.122	Low	98%	0.313	Low	87%	0.022	Low	77%	0.206	Low	55%	0.621
	High	70%		High	100%		High	62%		High	68%		High	60%	
CXCL5	Low	78%	0.381	Low	98%	0.844	Low	83%	0.53	Low	78%	0.531	Low	56%	0.629
	High	72%		High	100%		High	78%		High	68%		High	62%	
CXCL6	Low	78%	0.32	Low	77%	0.32	Low	78%	0.739	Low	73%	0.256	Low	61%	0.365
	High	72%		High	73%		High	82%		High	82%		High	47%	
CXCL7	Low	77%	0.828	Low	100%	0.054	Low	82%	0.274	Low	70%	0.748	Low	76%	0.045
	High	72%		High	92%		High	68%		High	77%		High	51%	
CXCL8	Low	78%	0.13	Low	98%	0.785	Low	82%	0.135	Low	78%	0.238	Low	59%	0.813
	High	68%		High	100%		High	68%		High	66%		High	52%	
SCORE		−2			−1			−5			−5			0	

**Table 3 cells-11-03933-t003:** C29 crosses the blood–brain barrier. The table shows the amounts of C29 (ng/mg cerebellum) that passed the blood–brain barrier in 10 cerebellar mice previously treated with 100 mg/kg and 10 others treated with 200 mg/kg. In each group, 5 mice were trafficked once, and their cerebellum was recovered after 3 h. Five mice were treated daily for 5 days (their cerebellum was recovered 24 h after the last treatment). C29 was determined by HPLC.

C29 Gavage	Acute (3 h)	Chronic (5 days)
100 mg/kg	ng C29/mg cerebellum	
Cerebellum 1	0	0
Cerebellum 2	0	0
Cerebellum 3	0	0
Cerebellum 4	0	0
Cerebellum 5	5.394	0
200 mg/kg	ng C29/mg cerebellum	
Cerebellum 1	13.1	0
Cerebellum 2	9.3	0
Cerebellum 3	7.4	0
Cerebellum 4	4.7	0
Cerebellum 5	6.4	0

## Data Availability

The data of Kaplan–Meyer curves and mRNA expression in MB patients were obtained from the R2: Genomics Analysis and Visualization Platform” (http://r2.amc.nl, accessed on 1 June 2019). The other row data and materials are available from the corresponding author Sonia Martial (sonia.martial@univ-cotedazur.fr, +33-4-92-03-12-29).

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
