# Peer review of "Targeting of the ELR+CXCL/CXCR1/2 Pathway Is a Relevant Strategy for the Treatment of Paediatric Medulloblastomas"

_cells, 2022, doi:10.3390/cells11233933_

Round 1

Reviewer 1 Report

In this article, authors characterize the impact of ELR+CXCL/CXCR1/2 targeting by a novel inhibitor C29, derived from SB225002, which has shown promise in mitigating cancer growth in the context of renal cancer. They start by demonstrating that the CXCR1/2 receptor and its partner ELR+ cytokine ligands, CXCL 1,2,3,5,6,7,8, are deregulated in medulloblastomas and that this deregulation is associated with poor prognosis. Here, the CXCR1/2 and CXCL7 deregulation are apparent when examining multiple MB datasets compared to control, while the connection to poor prognosis is established for CXCR2 in combined MB tumors. They then examine the expression of CXCR1/2 and the ELR+ CXCL cytokines in 2 SHH-type and 2 group 3-type MB cell lines, showing CXCR2 is deregulated in all cell lines and a smattering of CXCL cytokines is upregulated in specific cell lines, most notably CXCL1/2/8 in DAOY and CXCL5 in ONS-76, both SHH-type MB cell lines. Next, they begin their C9 characterization, first showing an IC50 in the 4 cell lines of <5uM at 72h and a propensity to cell death compared to normal at 5 uM. Using functional studies, i.e. colony formation, spheroid invasion assays, and neo-vessel formation, they evidence the efficacy of C29 in inhibiting these cancer cell functions in 2-3 cell lines. They follow up these studies with an analysis of ERK/AKT signaling linked to CXCR1/2 receptor activation. Finally, they show that C29 can inhibit cancer cell growth and function in radio-resistant DAOY and HDMB03 cells and in ONS-76 cells transplanted into healthy mouse cerebellar tissue ex vivo.

Overall, authors should be commended on the breadth of studies performed to characterize C29. They use multiple cell lines, examine several functional assays for cell growth and proliferation. They examine the impact of C29 on cancer cell growth ex vivo and  show efficacy of their inhibitor in radio-resistant cells. They somewhat reveal signaling perturbations induced by C29. 

Despite these strengths, some important weaknesses do persist that need to be addressed to ready their work for publication. These are delineated below: 

Research in context

1.     Line 39 and 45, please provide citations to back up these claims. 

2.     Line 51: cancer cells should be singular.

Introduction

1.     Line 76: The discussion on cure rates seems simplistic in that authors discuss MB as a whole, wherein in today’s reality, MBs are discussed in a subgroup-specific manner, unless authors want to discuss MB using average and high-risk designations. So, the cure rates should be qualified either for each subgroup or for average vs. high-risk, as MB is not a single disease, as the authors acknowledge, that can be thought of with a single cure rate. 

2.     Line 78: Please modify to: Medulloblastomas are a heterogenous group of tumors comprised of four primary molecular subgroups: 

3.     Remove “group” from Wingless and SHH; add “group” prior to 4. 

4.     Line 80: Add “respectively” after pathways. 

5.     Line 81: p53 inactivation is of low incidence in primary medulloblastomas; instead in reactivation/recurrence p53 inactivation is high. This is a defining difference between primary and recurrent MB. 

6.     Line 88: The 70% survival rate is for high-risk disease in the citation; it’s 85% in the Gajjar paper and 86% in the Parker paper for average-risk patients. 

7.     Line 91: Relapses are fatal but are infrequent for the WNT and SHH groups (virtually absent in WNT) and define the highly aggressive group 3 disease with the highest rates of recurrence. Refer to the Cavalli paper for further details and a table that shows the differences well. 

Results

1.     Supplementary Figure 2 does not show any association between CXCR1/2 and prognosis nor a reliable trend between any of the ELR+ cytokines and any subgroup. CXCL3 shows that high expression is problematic for SHH, CXCL1 for group 4; and CXCL7 high expression is potentially protective for group 3 disease. Moreover, expression of CXCR1/2 and the cytokines were shown to be deregulated in total MB tumors seen. The authors did not show subgroup-specific deregulation of any of these. So Line 276 is not supported by the data, i.e. “… poor prognosis in a subgroup-dependent manner.” In fact, the subgroup-specific data detracts from the main point of the paper. Suggest removing Suppl Figures 1 and 2 since they do not show any real significant connections; instead, authors can mention that subgroup-specific trends were not found and focus on MB as a whole as they have done in Figure 1. As a result, lines 246-279 will have to be revised. 

2.     For Figure 2, authors should compare the expression of receptors and cytokines to a normal like normal human astrocytes or neural progenitor cells. Comparing cell lines to each other seems of little value when the point is to show that these receptors/ligands are deregulated in cancer cells. Need to compare to a “normal” control cell line. Also, keep the designation of the x-asix consistent between figures, i.e. A and B have the cell line data arranged differently. Also, please consistently add significance values where indicated (figure 2B, R2, L3, L8). 

3.     Figure 3: Please add cell lines to the title of each IC50 graph; also, line 316, neither CXCR2 nor CXCL8 in SHH correlate with poor patient outcomes (Suppl Fig 2). In fact, in SHH, CXCL3 high expression is associated with poor prognosis, but this data has not been presented in vitro. Authors cannot conclude that CXCR1/2 or any of the cytokines are associated with poor prognosis in a subgroup-specific manner. They will be better served by being more general about the axis being deregulated in MB in general. 

4.     Figure 5: can authors also show the impact of C9 on apoptosis via WB (cleaved PARP/PARP, cleaved caspase 3/caspase3, Bcl, BAD/BAX, to suggest a few) or FACS (Annexin, PI)?  

Reviewer 2 Report

Penco-Campillo and colleagues present a novel study investigating the in vitro and ex vivo efficacy of a new chemokine receptor Cxcr1/Cxcr2 inhibitor, C29 in medulloblastoma (MB) cell lines.

Major Concerns:

1. Although the authors present a reasonably compelling argument using various in vitro and ex vivo assays, further work is necessary and this needs to be outlined in the revised Discussion. There is potential to extend this work to humans, but additional studies will be required using genetically engineered mouse models with intact immune systems, such as the Ptc1 heterozygote mouse as well as in orthotopic xenografts of human medulloblastoma cell lines in immunocompromised mice.

2. There does not appear to be a co-author who is a Paediatric Neuro-oncologist. Several incorrect statements about the clinical presentation, treatment and prognosis of children with medulloblastoma were made, especially in the Research in Context section and to a lesser extent in the Abstract and Introduction sections.

3.

a. Comments about C29 as a radiosensitizer are unwarranted without experimental evidence to support these statements (Discussion, line 537). This is an important future direction of this research.

b. Although the authors have published previously on stem cells, the statement on Line 551 is not backed up by any data presented in the current mechanism, so the sentence requires substantial revision.

4. The experiments would be more conclusive with the use of si/shRNA knockdown of Cxcr1 and Cxcr4 in addition to the pharmacologic inhibitor, C29, especially but not necessarily limited for Figure 5 where these results would be quite meaningful.

5. The IC50 dose of C29 is in the millimolar rather than nanomolar dose range. This has implications for its use in humans for off-target effects. 

Specific Concerns:

1. Abstract

a. Line 17: Delete "the evolution of".

b. Line 18: Add "usually" prior to "fatal". About 10-15% of relapsed MB can be salvaged in the clinic via various treatment pathways.

2. Research in context

a. Line 37: This statement is incorrect and should be rewritten. MB is the most common malignant brain tumour of childhood. It is not the most aggressive: diffuse midline glioma H3K27 altered has 0-1% survival and paediatric high grade gliomas and ATRT are also very aggressive.

b. Line 40 and Line 77: The authors should refer to the paper by Northcott in JCO in 2010 that first described the four MB molecular subgroups. A portion of patients in each subgroup can present with either leptomeningeal disease (LMD) or distal metastases, but some or many of these patients can be cured, depending on the subgroup. For example, WNT patients with M+ disease are usually cured. Relapse can be either local or with new/progressive distal metastases and/or LMD. Consider "relapse locally and/or with distal metastases" for Line 40.

3. Introduction

A recent paper in Pediatric Blood and Cancer demonstrated that the combination of irinotecan, temozolomide and bevacizumab was superior to irinotecan and temozolomide combination therapy in a randomized phase 2 study for children with relapsed MB. This publication should be added to the Introduction since bevacizumab is an anti-VEGF treatment strategy.

4. Materials and Methods

a. The reader should be directed to the supplement wherein additional methods are provided, such as the XTT assay (for example).

b. What was the age of the mice used to obtain cerebella? Cerebellar development in mice is relatively delayed compared to human, but studies using juvenile mice may have been informative as well as adult mice.

c. Lines 202-211: Use the past tense only. There is use of mixed past and present tense as written.

d. Line 213: Provide a one sentence justification for the use of the R2 platform.

5. Results

a. Figure 1: Is there any data for human foetal, infant or child brain. Was the normal brain (NB) data only from adults? 

b. Figure 1c: Another database should be checked to validate the Cavalli database.

c. Table 1: For the methodology used to calculate the scores, has this methodology been used by other groups and validated? If yes, please provide a reference. If no, then please be less conclusive.

d. Figure 3: The term "metabolism" is very general. Please explain what you mean for the reader. For the XTT assay, what aspects of metabolism are measured? Be more specific. 

e. Figure 3: The primary mouse astrocytes (C8D1A) and the term "neural lineage" may confuse the reader, since astrocytes are not oligodendrocytes, interneurons or projection neurons. Use "nervous system lineage" instead (Line 352).

f. Figure 6: Explain "mesh" for the reader.

Minor Concerns:

Line 21: Change "children" to "childhood".

Line 31: Change "radiotherapy" to "therapy" since children receive both radiation and chemotherapy.

Line 37: Change "the" to "a". There are at least 3 other common tumours involving the cerebellum/posterior fossa, including ATRT, pilocytic astrocytoma and ependymoma.

Line 44: Define "ELR" first use either here or in the abstract. Not all readers are familiar with chemokine terminology.

Line 48: Use another word instead of "anergy" which usually means no response when presented with an antigen challenge.

Line 51: Change "cells" to "cell".

Line 68: Replace "responsible" with "associated".

Line 75: Place "up to" prior to "25%".

Line 80: Add ", respectively" after "pathways".

Line 87: Replace "X-rays" with "photons and increasingly with protons".

Line 91: Add "usually" prior to "fatal".

Line 175: Replace "killed" with "sacrificed".

Line 182: Replace "the mice" with "mouse".

Line 201: Change "follow" with "follows".

Line 360: Replace "before" with "prior".

Line 422: Delete "MB" after "reduced". It is unintentionally used twice in the sentence.

Line 436: Spell "scale" correctly.

Other: There is actually no Figure 2D yet the figure legend and text refer to it.

Round 2

Reviewer 2 Report

Overall, the authors have done a nice job responding to the reviewers' concerns. The revised manuscript is considerably improved. It is unfortunate that the knockdown experiments are not yet successful, but this is an important step to validate the results obtained from the C29 compound. Hence, a re-revised Discussion should specifically address this limitation.

Author Response

Again, we thank the reviewer who took even more time to read and comment our manuscript.

We acknowledge that the point raised by reviewer 2 is very important. Indeed, obtaining CXCR2 knockdown medulloblastoma models would add high value to our demonstration. Since, as previously mentioned, we were unable to generate knockdown models, using several protocols, as well as several commercial and validated siRNAs, we had to abandon this part of the project. However, as requested by the reviewer, this necessary set of experiments and its implications is now commented in the discussion (lines 550 - 568).